# Dynamic Reduction-Based Virtual Models for Digital Twins—A Comparative Study

**Soumya Maulik** [1,2,3,4,*] , **Daniel Riordan** [1,2,3,4] **and Joseph Walsh** [1,2,3,4]

1 IMaR Research Centre, Munster Technological University, V92 CX88 Tralee, Ireland; daniel.riordan@mtu.ie (D.R.); joseph.walsh@mtu.ie (J.W.)
2 School of Science Technology, Engineering and Mathematics, Munster Technological University, V92 CX88 Tralee, Ireland
3 Department of Technology, Engineering, and Mathematics, Munster Technological University, V92 CX88 Tralee, Ireland
4 Lero—Science Foundation Ireland Research Centre for Software, V92 NYD3 Limerick, Ireland
* Correspondence: Soumya.Maulik@research.ittralee.ie

**Abstract:** Digital models are the foundation of digital twins, which form the basis of autonomous off-road vehicles. Developing virtual models of off-road vehicles using dynamic reduction techniques is one of several approaches. The article commences with a comprehensive overview of the most widely used dynamic reduction methods and then introduces performance metrics for assessing their efficacies in the context of digital twins. The paper additionally includes a detailed mathematical derivation of the state-space representation for reduced-order finite element models. The state-space representation of the reduced finite element models facilitates their export to problem-solving environments for dynamic analysis. The state-space models are eventually solved utilizing the built-in libraries of numerical solvers in textual and graphical programming platforms. In addition, the article identifies the set of solvers that best suit the simulation of virtual models for off-road vehicles. This article also includes an evaluation of the simulation results for digital models with modes ranging from 0 to 30 Hz. In addition, the article demonstrates the lower bound of the frequency range necessary and sufficient to be retained in off-road vehicle virtual models. Finally, the paper presents the simulation outcomes for digital models of commercial off-road vehicles with custom-built virtual modules of powertrain, electrical, and control systems in a problem-solving environment.

**Keywords:** digital twin; industry 4.0; MATLAB; ANSYS; simulation; crane; modal analysis; dynamic substructuring; dynamic reduction; component modal synthesis

## 1. Introduction

The technical cornerstone of Industry 4.0 [1–19] is the Internet of Things (IoT) [20,21], which envisions integrating electronics, software, sensors, and network connectivity into devices to facilitate the seamless transfer of data over the internet. The introduction of IPV6 [22,23], the availability of affordable sensors, and the enhancement of computing hardware have eased data acquisition and the processing of sensory information. Consequently, data-driven decision-making and device management in the industrial environment are now more practically feasible than before. The decision-making and control of operating structures can be accomplished by comparing the pooled data from the sensors installed on them with the simulated output of their digital equivalents, a concept referred to as a "digital twin" [24–50], which is also one of the theoretical frameworks associated with the Industry 4.0 revolution. The use of digital twins for aviation prognostics and diagnostics operations has yielded considerable benefits for aerospace and space organizations. In the realm of the aerospace industry, NASA's initial definition of a digital twin was "an integrated multi-physics, multi-scale, probabilistic simulation of a vehicle or system that uses the best available physical models, sensor updates, fleet history, etc., to mirror the life of its flying twin. The digital twin is ultra-realistic and may consider one or more important and

interdependent vehicle systems" [51]. As stated in the definition, the primary objective of a digital twin is to replicate the life of an aircraft in cyberspace using the "best" physics-based digital model that consists of the integration of sub-components that represent the overall structure. While the aerospace industry and space agencies, in particular, have reaped immense benefits, the adoption of digital twins for heavy off-road vehicles is rare. There are potential benefits of digital twin models for the latter sector, including but not limited to enhanced operational efficiency, improved maintenance regimes that can prevent expensive faults, optimized asset utilization, reduced operating expenses, and remote commissioning. A virtual model that encompasses the entire set of coordinates of a full-order model for a structure accurately and consistently represents the deformed shape, but it requires solving large matrices. Consequently, simulation of virtual models with the entire set of principal coordinates is computationally expensive and has a prolonged runtime, preventing real-time synchronization with the activities of their physical counterparts operating in an industrial environment. This article proposes dynamic reduction-based virtual models as a solution to these challenges.

Dynamic reduction techniques came into existence from 1960 onwards and are referred to as "component modal synthesis" in the field of structural dynamics [52,53], and "substructuring" in statics [54], with the original objectives of (i) minimizing the computational cost of analysis of structures with a large number of degrees of freedom and (ii) analyzing constituent substructures of a structure independently in such a manner that it would be feasible to amalgamate the results. Nonetheless, dynamic model reduction schemes have the potential to make a substantial contribution to the development of physics-based virtual models.

Substructure coupling approaches may be primarily categorized based on the types of modes employed for modeling the reduced finite element representation of a component as

1. Fixed-Interface methods [55–57];
2. Free-Interface methods [58–61];
3. Loaded-Interface [62];
4. Hybrid methods [63,64].

The substructure interfaces in fixed-interface schemes are entirely constrained, whereas the substructure interfaces in free-interface methods are unrestricted. The "loaded-interface" methods, which augment the interface loading components in the structural matrices of the substructures, are related to the free-interface schemes. Hybrid techniques allow for arbitrary restrictions at substructure boundaries, and in the most generalized situation, these constraints can be a combination of the three types of outlined coupling methods. The fundamental objective of this research is to identify the high-fidelity dynamic model reduction approach that "best" suits the development of digital models for off-road vehicles. In this article, Craig–Bampton and Hintz's modal synthesis techniques for fixed-interface and free-interface methods, respectively, are considered.

In addition to reduced structural models, digital models for off-road vehicles comprise virtual models of driveline systems, electrical components, and powertrain control modules. Textual and graphical programming platforms, unlike FEA-based platforms, use uncluttered programming syntax and are thus deemed appropriate for the design, modeling, and simulation of specialized powertrain, electrical, and control systems. Finally, the state-space representation of reduced structural models can be incorporated into problem-solving environments and subsequently coupled with custom-built powertrain, electrical, and control systems to develop virtual models for off-vehicles in cyberspace.

The virtual models facilitate the yield of physics-based digital twins for off-road vehicles; however, there is hardly any extant literature presenting their development methods. This paper proposes dynamic reduction methods for developing reduced-order models that subsequently represent the virtual models for off-road vehicles. Simulation of these digital models with reduced degrees of freedom improves speed without impacting accuracy. Furthermore, built-in packages of FEA-based software, predominately used for structural modeling and analysis, incorporate these high-fidelity dynamic reduction techniques,

thereby enabling the development of the reduced-order models on the same platforms where designed. This article outlines the state-space representation method for these reduced-order models. The state-space models facilitate the simulation of reduced-order models in problem-solving environments. This paper, in addition, provides a comprehensive mathematical derivation for the state-space representation of reduced-order models. To accomplish real-time synchronization with the activities of a physical counterpart in an industrial environment, the simulation time of virtual models must be less than or equal to the operating time of physical equivalents. As a result, a comparative analysis of the execution times in a problem-solving environment of digital models developed utilizing Craig–Bampton and Hintz's component modal synthesis is provided. It is worth noting that these virtual models are eventually solved utilizing solvers that are available as built-in libraries on textual and graphical programming platforms; hence, selecting the optimal solver among the available solvers is critical. Consequently, this paper presents a comparative assessment of the execution times of virtual models for off-road vehicles employing these solvers.

The remainder of this article is structured as follows: Section 2 presents a comprehensive review of the dynamic reduction techniques and the state-space representation of the reduced finite-element model. Section 3 outlines the metrics for evaluating the dynamic reduction methods. Section 4 presents the results, and finally, Section 5 concludes the article.

## 2. Component Modal Synthesis

The superposition of principal modes, also referred to as the normal mode method [65], is a fundamental and extensively used analytical approach for solving vibration problems. In this method, linear transformations in terms of modal vectors, in conjunction with the orthogonality of modal vectors, enable the transformation of a set of simultaneous equations of motion in physical coordinates into a set of independent modal equations. For large and complex industrial structures, an eigensolution, which is the most fundamental part of a dynamic analysis using the normal mode method, becomes uneconomical due to the sheer number of equations of motion. Dynamic reduction methods alleviate the computational constraints of dealing with large matrices by obtaining the principal modes of the overall system from parts rather than as a whole. In these methods, multiple eigenvalue problems involving individual constituent components represented by smaller matrices are used rather than a single eigenvalue problem representing the entire structure. The coordinate system obtained by synthesizing the components is assembled for system synthesis, yielding one final eigenvalue problem, typically of smaller size.

A substructure, in general, contains a set of constraints at its interfaces coupled with neighboring substructures. General displacement within a substructure is defined in dynamic reduction methods by superimposing displacements relative to component boundaries, from which a set of generalized coordinates applicable to the entire structure is synthesized. One of the subcategories of dynamic reduction methods is fixed-interface component modal synthesis. The fixed-interface component modal synthesis by Craig–Bampton is one of the most widely used dynamic reduction techniques, and most FEA-based systems have built-in libraries, as outlined in the following section.

### 2.1. Fixed-Interface Modal Synthesis Technique

Hurty pioneered the notion of fixed-interface component modal synthesis [56], but the method was incompatible with automation. Craig–Bampton [57] modified Hurty's approach, removing the limitations that facilitated the implementation of the modal synthesis process on computing platforms. Craig–Bampton's method analyzes components by the superimposition of constraint and fixed-constraint normal modes.

Constraint modes are the displacements of interior coordinates in a substructure induced by sequential unit displacements of interface constraints, with the remaining boundary constraints being constrained. The boundary constraints are usually imposed on the coupled points between interconnected substructures. These coupled points are

subjected to unit displacement in the direction of each degree of freedom, considered one at a time, resulting in the deflection of the substructure. The deflected configurations of the substructure are the constraint modes for that individual substructure. For the *i*th substructure, the linear relationship between the externally applied force and stiffness for static analysis is given by

$$\left\{ \begin{array}{c} p_B \\ p_I \end{array} \right\}_i = \left[ \begin{array}{c:c} k_{BB} & k_{BI} \\ \hdashline k_{BI}^\mathsf{T} & k_{II} \end{array} \right]_i \left\{ \begin{array}{c} u_B \\ u_I \end{array} \right\}_i \tag{1}$$

Because there is no external force acting on the unconstrained coordinates in constraint mode analysis, taking the bottom partition of Equation (1) yields

$$[k_{BI}]_i^\mathsf{T} \{u_B\}_i + [k_{II}]_i \{u_I\}_i = \{0\}$$
$$\{u_I\}_i = -[k_{II}]_i^{-1}[k_{BI}]_i^\mathsf{T} \{u_B\}_i = [\phi_c]_i \{u_B\}_i \tag{2}$$

Fixed-constraint normal modes are defined by displacements of interior points in the component relative to the interface constraints. These are the normal modes of vibration with all interface constraints fixed, typically computed by eigenvalue analysis of the substructure. The zero-input response and eigenanalysis of a system are analogous, and therefore, in the absence of external excitation in the internal degrees of freedom, the undamped equation of motion for the *i*th substructure can be expressed as,

$$\left[ \begin{array}{c:c} m_{BB} & m_{BI} \\ \hdashline m_{BI}^\mathsf{T} & m_{II} \end{array} \right]_i \left\{ \begin{array}{c} \ddot{u}_B \\ \ddot{u}_I \end{array} \right\}_i$$
$$+ \left[ \begin{array}{c:c} k_{BB} & k_{BI} \\ \hdashline k_{BI}^\mathsf{T} & k_{II} \end{array} \right]_i \left\{ \begin{array}{c} u_B \\ u_I \end{array} \right\}_i = \left\{ \begin{array}{c} p_B \\ 0 \end{array} \right\}_i \tag{3}$$

In fixed-interface normal mode analysis, the constraints on the interface coordinates for a substructure are constrained, resulting in

$$\{u_B\} = 0$$

and

$$\{\ddot{u}_B\} = 0 \tag{4}$$

Substituting Equation (4) into Equation (3) and using the lower partition yields

$$[m_{II}]_i \{\ddot{u}_I\}_i + [k_{II}]_i \{u_I\}_i = \{0\} \tag{5}$$

The free vibration solution for Equation (5) is of the form $\{u_I\}_i = [\phi_n]_i \{\eta_n\}_i$. Therefore, Equation (5) reduces to the eigenvalue problem

$$([k_{II}]_i - [\omega_n^2]_i [m_{II}]_i)\{\phi_I\}_i = 0$$

and

$$\left| [k_{II}]_i - [\omega_n^2]_i [m_{II}]_i \right| = 0 \tag{6}$$

where $[\omega_n^2]_i = diag(\omega_{n_1}^2, \ldots, \omega_{n_{N_i^I}}^2)$. The mass normalized eigenvectors form the respective columns of $[\phi_n]_i$ of the *i*th constrained substructure. The elements of $[\phi_n]_i$ to be retained are then collected in $[\bar{\phi}_n]_i$ and thereby retain the associated interior degrees of freedom of the substructure. The interface coordinates being entirely constrained in fixed-constraint normal mode analysis renders the upper partition of Equation (3) redundant. It is worth mentioning that the constraint modes represent the static characteristics of the substructure, while the fixed constraint normal modes, as determined by vibration analysis, demonstrate the dynamic behavior of the substructure.

Finally, the transformation from physical coordinates to generalized coordinates for the $i$th substructure is then

$$
\begin{aligned}
\{u\}_i &= \left[\begin{array}{c:c} I & 0 \\ \hdashline \phi_c & \phi_n \end{array}\right]_i \left\{\begin{array}{c} u_B \\ \hdashline \eta_n \end{array}\right\}_i \\
&= [T_{CB}]_i \left\{\begin{array}{c} u_B \\ \hdashline \eta_n \end{array}\right\}_i
\end{aligned}
\tag{7}
$$

It is worth noting that truncating the constraint modes is not feasible since they characterize the entire motion of the system.

The reduced mass and stiffness matrix of the $i$th substructure in generalized coordinates is then expressed as

$$
\begin{aligned}
[\bar{m}]_i &= [T_{CB}]_i^{\mathsf{T}} [m]_i [T_{CB}]_i \\
[\bar{k}]_i &= [T_{CB}]_i^{\mathsf{T}} [k]_i [T_{CB}]_i
\end{aligned}
\tag{8}
$$

The equations of motion for the overall structure in the unconnected form are eventually obtained by grouping together the equations of motion for all $\tau \in \mathbb{N}^+$ substructures. A compatibility matrix ($[\beta]$) renders constraints on the interconnected substructures, ensuring compatible displacements at substructure boundaries. The boundary coordinates for any two adjoining substructures $i$ and $j$ are expressed in their respective local coordinate frames of reference, $\{u\}_i$ and $\{u\}_j$, and consequently, it is imperative to express their frame of reference in a shared global reference frame, $\{\bar{u}\}$, prior to establishing compatible displacement connections. Let the boundary coordinates for the $i$th and $j$th substructures in the global frame of reference be $\{\bar{u}_B\}_i^s$ and $\{\bar{u}_B\}_j^s$, respectively, at a common connection point $s$. Then, the following necessary and sufficient conditions must be satisfied at the boundary to ensure that a substructure's displacements on its boundary correspond to those of its adjoining substructures in the global frame of reference:

$$
\{\bar{u}_B\}_i^s = \{\bar{u}_B\}_j^s
\tag{9}
$$

For the sake of brevity, superscript ($s$) in Equation (9) will be omitted. In terms of interconnected substructures, there are no compatibility constraints for the modal coordinates. Furthermore, any arbitrary substructure $i$ with multiple couplings to adjoining substructures may be expressed using a matrix notation $[\beta]_i$. Finally, the general form of the transformation of coordinates to represent the overall structure is as follows:

$$
\left\{\begin{array}{c} u_{B_1} \\ \eta_{n_1} \\ \vdots \\ u_{B_\tau} \\ \eta_{n_\tau} \end{array}\right\} = \left[\begin{array}{c} \beta_1 \\ \vdots \\ \beta_\tau \end{array}\right] \left\{\begin{array}{c} \bar{u}_{B_1} \\ \vdots \\ \bar{u}_{B_{\tau_c}} \\ \bar{u}_B \\ \eta_{n_1} \\ \vdots \\ \eta_{n_\tau} \end{array}\right\}
= [\beta]\{\mu\}
\tag{10}
$$

The uncoupled set of boundary coordinates mentioned in Equation (10) is where external excitations are applied. Finally, the equation of motion for free vibration of the complete structure may be expressed as

$$
[M_\mu]\{\ddot{\mu}\} + [K_\mu]\{\mu\} = 0
\tag{11}
$$

where $[M_\mu] = [\beta]^\mathsf{T}[\bar{M}_\mu][\beta]$ and $[K_\mu] = [\beta]^\mathsf{T}[\bar{K}_\mu][\beta]$, and where

$$
\bar{M}_\mu = \begin{bmatrix} [\bar{m}]_1 & & \\ & \ddots & \\ & & [\bar{m}]_\tau \end{bmatrix}, \quad \bar{K}_\mu = \begin{bmatrix} [\bar{k}]_1 & & \\ & \ddots & \\ & & [\bar{k}]_\tau \end{bmatrix}
$$

The Craig–Bampton modal synthesis technique has a high degree of accuracy, and built-in libraries are usually available in most FEA-based software products [66–68].

### 2.2. Free-Interface Modal Synthesis Techniques

The presence of rigid body motion is one of the characteristics of free–free or unconstrained systems. For such systems, the stiffness matrix is singular, and hence no flexibility matrix exists, jeopardizing static analysis. In his modal synthesis approach, Hintz eliminated the rigid body modes using the inertia relief matrix, facilitating the analysis of the elastic behavior of such unconstrained systems. The free-interface modal synthesis scheme proposed by Hintz [61] for free–free systems constitutes attachment modes for unconstrained systems or inertia relief modes, constraint modes, and free-interface normal modes of substructures.

For the $i$th substructure, the free-interface normal modes are obtained by setting Equation (3) for $\{p_B\} = 0$. By substituting $\{p_B\} = 0$, Equation (3) reduces to

$$
\begin{bmatrix} m_{BB} & m_{BI} \\ m_{BI}^\mathsf{T} & m_{II} \end{bmatrix}_i \begin{Bmatrix} \ddot{u}_B \\ \ddot{u}_I \end{Bmatrix}_i \\
+ \begin{bmatrix} k_{BB} & k_{BI} \\ k_{BI}^\mathsf{T} & k_{II} \end{bmatrix}_i \begin{Bmatrix} u_B \\ u_I \end{Bmatrix}_i = \begin{Bmatrix} 0 \\ 0 \end{Bmatrix}_i
\tag{12}
$$

Thus,

$$
([k]_i - [\omega_{fn}^2]_i[m]_i) \begin{Bmatrix} \phi_B \\ \phi_I \end{Bmatrix}_i = \{0\}
$$

and

$$
\left| [k]_i - [\omega_{fn}^2]_i[m]_i \right| = 0
\tag{13}
$$

where $[\omega_{fn}^2]_i = diag(\omega_{fn_1}^2, \dots, \omega_{fn_{N_i}}^2)$. If the $i$th substructure is unrestrained, then rigid body modes are included in $[\omega_{fn}^2]_i$. The mass normalized eigenvectors corresponding to the free-interface normal modes belonging to the interior and boundary degrees of freedom for the $i$th substructure to be retained are gathered in $[\bar{\phi}_{In}]_i$ and $[\bar{\phi}_{Bn}]_i$, respectively, thereby retaining the corresponding coordinates. Furthermore, truncation of the free-normal modes renders elimination of the associated degrees of freedom.

Let $W_i$ be the set of physical coordinates for the $i$th substructure, where $W_i \subset N_i \setminus O_i$, and where $O_i$ is the statically determinate constraint set sufficient to provide restraint against rigid body motion. Furthermore, let us define a set such that $L_i \subset W_i$. The static deflection as a consequence of an applied unit force on $w_j^i \in W_i$, where $j \in \{1, \dots, |W|\}$ while the remaining of the degrees of freedom in the set $W_i$ are devoid of force, is characterized as the attachment mode.

Separating the displacement vector for the $i$th substructure into pure rigid body displacement $\{u_o\}$ and elastic deformation vector $\{u_e\}$ yields

$$
\{u\}_i = \{u_o\}_i + \{u_e\}_i
\tag{14}
$$

The displacements in physical coordinates can be expressed in modal coordinates in the following way

$$
\{u\}_i = \begin{bmatrix} \phi_o & \vdots & \phi_e \end{bmatrix}_i \left\{ \frac{\eta_o}{\eta_e} \right\}_i \tag{15}
$$

Using the non-homogeneous undamped equation of motion and knowing that $[k]_i[\phi_o]_i = 0$, the modal coordinates associated with rigid body motions can be explicitly represented as follows

$$
[m]_i[\phi_o]_i\{\ddot{\eta}_o\}_i = \{p\}_i \tag{16}
$$

Premultiplying Equation (16) by $[\phi_o]_i^\mathsf{T}$ yields

$$
\{\ddot{\eta}_o\}_i = [\phi_o]_i^\mathsf{T}\{p\}_i \tag{17}
$$

where $[\phi_o]_i^\mathsf{T}[m]_i[\phi_o]_i = [I]$.

Suppose $\{p_o\}_i$ is the D'Alembert interia forces that are in equilibrium with the inertia loads, and therefore

$$
\{p_o\}_i + [m]_i\{\ddot{u}_o\}_i = 0 \tag{18}
$$

Equation (18) can be modified further using Equation (17), which yields

$$
\begin{aligned}
\{p_o\}_i &= -[m]_i\{\ddot{u}_o\}_i \\
&= -[m]_i[\phi_o]_i\{\ddot{\eta}_o\}_i \\
&= -[m]_i[\phi_o]_i[\phi_o]_i^\mathsf{T}\{p\}_i
\end{aligned} \tag{19}
$$

The equilibrated load system $\{p_e\}_i$ for the $i$th component can be represented as

$$
\begin{aligned}
\{p_e\}_i &= \{p\}_i + \{p_o\}_i \\
&= \{p\}_i - [m]_i[\phi_o]_i[\phi_o]_i^\mathsf{T}\{p\}_i \\
&= [\Psi]_i\{p\}_i
\end{aligned} \tag{20}
$$

where $[\Psi]_i = [I] - [m]_i[\phi_o]_i[\phi_o]_i^\mathsf{T}$ is the inertia-relief matrix.

The attachment modes ($[\phi_a]_i$) for the $i$th unrestrained component relative to the $O$ constraints can be expressed as

$$
\begin{bmatrix} k_{ww} & k_{wl} & \vdots & k_{wo} \\ k_{lw} & k_{ll} & \vdots & k_{lo} \\ \hline k_{ow} & k_{ol} & \vdots & k_{oo} \end{bmatrix}_i \begin{bmatrix} \phi_w^a \\ \phi_l^a \\ 0 \end{bmatrix}_i = [\Psi]_i \begin{bmatrix} I \\ 0 \\ 0 \end{bmatrix}_i \tag{21}
$$

Finally, the attachment modes are deduced from Equation (21) using the top-left partition of the stiffness matrix.

The constraint modes are assessed in the same manner as described in the preceding section. It is important to note that both attachment and constraint modes are instances of static response modes that manifest from constant external load and displacements, respectively. For the $i$th substructure, Hintz's transformation matrix for dynamic analysis is represented by

$$
[T_H]_i = \begin{bmatrix} I & \vdots & 0 & \vdots & \overline{\phi}_{Bn} \\ \hline \phi_c & \vdots & \phi_a & \vdots & \phi_{In} \end{bmatrix}_i \tag{22}
$$

The reduced mass and stiffness matrix for the $i$th substructure can now be expressed as

$$
\begin{aligned}
[\bar{m}]_i &= [T_H]_i^\mathsf{T} [m]_i [T_H]_i \\
[\bar{k}]_i &= [T_H]_i^\mathsf{T} [k]_i [T_H]_i
\end{aligned} \tag{23}
$$

The constraints of interconnected substructures are implemented using a transformation matrix $[\beta]$. The final reduced structural mass and stiffness matrix can be represented as

$$
\begin{aligned}
[M_\mu] &= [\beta]^\mathsf{T} [\bar{M}_\mu][\beta] \\
[K_\mu] &= [\beta]^\mathsf{T} [\bar{M}_\mu][\beta]
\end{aligned} \tag{24}
$$

where

$$
[\bar{M}_\mu] = \begin{bmatrix} \bar{m}_1 & & \\ & \ddots & \\ & & \bar{m}_\tau \end{bmatrix}, \quad [\bar{K}_\mu] = \begin{bmatrix} \bar{k}_1 & & \\ & \ddots & \\ & & \bar{k}_\tau \end{bmatrix}
$$

Hintz's method for assessing free–free systems such as launch vehicles, aircraft, and spacecraft is very efficient and is prevalently used by aerospace and space organizations.

### 2.3. State-Space Representation of Reduced-Order Model

The reduced structural matrices are stated in generalized coordinates in both the free-interface and fixed-interface component modal synthesis, as shown by Equations (11) and (24). Mode superposition and mode acceleration [69] both facilitate the transformation of modal displacements to physical displacements for a structure. In practice, during the solution phase of the modal synthesis procedures, the FEA-based software writes and stores the assembled reduced structural matrices and associated degrees of freedom in files. For instance, Ansys [66] stores this structural information in a binary file labeled "full." PyAnsys [70] is a free and open-source Python interface for Ansys that enables the retrieval and import of Ansys structural data into Python. It is desirable to express the reduced structural matrices obtained from FEA-based software in a platform-independent format. The present research considers state-space representation [71–73]. Expressing physical systems in a state-space representation is a well-established approach in control engineering; thus, most software platforms for modeling and simulating multi-domain dynamical systems include state-space blocks. The remainder of this section aims to establish a generalized state-space representation for the digital model of a reduced finite element model, and this approach applies to any arbitrary off-road vehicle.

For a reduced system, the undamped equation of motion can be expressed as

$$
[M_r]\{\ddot{u}_r\} + [K_r]\{u_r\} = \{p_r\} \tag{25}
$$

Although the reduced mass and stiffness matrices are symmetric, the non-zero off-diagonal elements render Equation (25) a system of coupled differential equations, which is computationally expensive to solve. In order to solve a system of coupled differential equations, the inertial and elastic decoupling of Equation (25) is essential, which is accomplished using the normal mode method. The equation of motion for the free vibration of the undamped reduced system is expressed as

$$
[M_r]\{\ddot{u}_r\} + [K_r]\{u_r\} = \{0\} \tag{26}
$$

which is homogeneous. The solution of Equation (26) is of the form

$$
\{u_r\} = [\phi]\{\eta_r\} \tag{27}
$$

The modal matrix $[\phi]$ has the following orthogonality features in relation to the mass and stiffness matrices:

$$[\phi]^{\mathsf{T}}[M_r][\phi] = [I]$$
$$[\phi]^{\mathsf{T}}[K_r][\phi] = [\omega_*^2] \tag{28}$$

where $[\omega_*^2] = diag(\omega_{*1}^2, \ldots, \omega_{*R}^2)$. The retained set of modes in the reduced-order model is a subset of the natural frequencies of the full-order finite element model. Mathematically, the relationship can be expressed as

$$\{\omega_*\} \subset \{\omega\} \tag{29}$$

where

$$\omega_{*i} \approx \omega_j \ : \ i \in \{1,\ldots,R\}, j \in \{1,\ldots,N\}$$

Thus, the coupled equations of motion in Equation (25) can be expressed as uncoupled equations of motion in modal coordinates using the orthogonality relations of the modal matrix, as

$$\{\ddot{\eta}_r\} + [\omega_*^2][\{\eta_r\}] = [\phi]^{\mathsf{T}}\{p_r\} \tag{30}$$

Let us define $\eta_1, \dot{\eta}_1, \ldots, \eta_R, \dot{\eta}_R$ as the state variables, and for convenience, let us rename the state variables as $x_1, x_2, \ldots x_{2R}$, where

$$
\begin{aligned}
x_1 &= \eta_1 \\
x_2 &= \dot{\eta}_1 \\
&\vdots \\
x_{2R-1} &= \eta_R \\
x_{2R} &= \dot{\eta}_R
\end{aligned} \tag{31}
$$

Equation (31) can be expressed in the following manner

$$
\begin{aligned}
\dot{x}_1 &= x_2 \\
\dot{x}_2 &= \ddot{x}_2 \\
&\vdots \\
\dot{x}_{2R-1} &= x_{2R} \\
\dot{x}_{2R} &= \ddot{x}_{2R}
\end{aligned} \tag{32}
$$

Using Equations (30)–(32), the equation of motion in principal coordinates can be expressed in terms of the state variables as

$$
\begin{Bmatrix} \dot{x}_1 \\ \dot{x}_2 \\ \vdots \\ \dot{x}_{2R-1} \\ \dot{x}_{2R} \end{Bmatrix} =
\begin{bmatrix}
0 & 1 & \cdots & 0 & 0 \\
-\omega_{*1}^2 & 0 & \cdots & 0 & 0 \\
\vdots & \vdots & \vdots & \vdots & \vdots \\
0 & 0 & \cdots & 0 & 1 \\
0 & 0 & \cdots & -\omega_{*R}^2 & 0
\end{bmatrix}
\begin{Bmatrix} x_1 \\ x_2 \\ \vdots \\ x_{2R-1} \\ x_{2R} \end{Bmatrix}
$$
$$
+ [\phi]^{\mathsf{T}} \begin{Bmatrix} 0 \\ p_1 \\ \vdots \\ 0 \\ p_R \end{Bmatrix} \tag{33}
$$

The modal displacements and velocities are obtained by solving Equation (33). For example, $x_1$, $x_2$ represent the modal displacement and velocity of the first retained degree

of freedom. Let $D$ be the number of degrees of freedom where the structure's displacement and velocity are to be monitored. Then, suppose $(S_i^D | i \in \{1, \ldots, D\})$ and $(S_j | j \in \{1, \ldots, R\})$ are the sequence of the set of degrees of freedom where the kinematics to be observed and the sequence of the set of retained degrees of freedom in the reduced structure, respectively. Let us now define a matrix $[\bar{C}]_{2D \times 2R}$ in the following way:

$$
\begin{aligned}
\bar{C}_{2i-1,2j+1} &= 1 : S_i^D = S_j, \forall i \in \{1, \ldots, D\}, \forall j \in \{1, \ldots, R\} \\
\bar{C}_{2i,2j+2} &= 1 : S_i^D = S_j, \forall i \in \{1, \ldots, D\}, \forall j \in \{1, \ldots, R\} \\
\bar{C}_{i,j} &= 0 : \text{otherwise}
\end{aligned}
\tag{34}
$$

Let us define $\Lambda^{ss}$ and $B^{ss}$ as follows:

$$
\Lambda^{ss} = \{\lambda^{ss} \in \mathbb{N}^+ | \lambda^{ss} \text{ is even and } \lambda^{ss} \leq 2R\}
$$

and

$$
B^{ss} = \{\beta^{ss} \in \mathbb{N}^+ | \beta^{ss} \text{ is even and } \beta^{ss} \leq 2R\}
$$

The original modal matrix $[\phi]$ of the reduced structure can then be modified as follows:

$$
\phi_{i,j}^{ss} = \begin{cases} \phi_{\frac{i}{2},\frac{j}{2}} & : \forall i = \lambda^{ss} \in \Lambda^{ss}, \forall j = \beta^{ss} \in B^{ss} \\ 0 & : \text{otherwise} \end{cases}
\tag{35}
$$

where $[\phi^{ss}]$ is the reduced structure's modified modal matrix. The necessity for the aforementioned modification of the modal matrix is to accommodate the additional states related to the system velocity. The kinematic vector at selected degrees of freedom in physical coordinates can be derived using Equations (27), (34), and (35), yielding

$$
\{y\} = [C]\{x\}
\tag{36}
$$

where $[C] = [\bar{C}][\phi^{ss}]$, and $\{x\}$ is the system kinematics vector in modal coordinates.

Furthermore, only a subset of the retained degrees of freedom in a reduced finite element model of a structure is subjected to external excitation. Suppose $r_{in}$ is the number of excitations applied at distinct degrees of freedom of the structure, where $r_{in} \leq R$. Let us further assume $(S_i^{r_{in}} | i \in \{1, \ldots, r_{in}\})$ is the sequence of the set of degrees of freedom where the external excitations are applied. As a result, Equation (33) can be rewritten as

$$
\begin{Bmatrix} \dot{x}_1 \\ \dot{x}_2 \\ \vdots \\ \dot{x}_{2R-1} \\ \dot{x}_{2R} \end{Bmatrix} = \begin{bmatrix} 0 & 1 & \ldots & 0 & 0 \\ -\omega_{*1}^2 & 0 & \ldots & 0 & 0 \\ \vdots & \vdots & \vdots & \vdots & \vdots \\ 0 & 0 & \ldots & 0 & 1 \\ 0 & 0 & \ldots & -\omega_{*R}^2 & 0 \end{bmatrix} \begin{Bmatrix} x_1 \\ x_2 \\ \vdots \\ x_{2R-1} \\ x_{2R} \end{Bmatrix}
$$
$$
+ [\phi^{ss}]_{2R \times 2R} [\beta^p]_{2R \times r_{in}} \begin{Bmatrix} |p_1| \\ \vdots \\ |p_{r_{in}}| \end{Bmatrix}_{r_{in} \times 1}
\tag{37}
$$

where $|p_j| \; \forall j \in \{1, \ldots, r_{in}\}$ is the magnitude of $j$th externally applied excitation in physical coordinates, and $[\beta^p]$ is a Boolean matrix defined as follows

$$
\begin{aligned}
\beta_{2i,j}^p &= 1 : S_j^{r_{in}} = S_i, \forall i \in \{1, \ldots, R\}, \forall j \in \{1, \ldots, r_{in}\} \\
\beta_{i,j}^p &= 0 : \text{otherwise}
\end{aligned}
\tag{38}
$$

Finally, the state-space representation of a reduced finite element model can be expressed using Equation (37) as

$$\begin{aligned}\{\dot{x}\} &= [A]\{x\} + [B]\{z\} \\ \{y\} &= [C]\{x\} + [D]\{z\}\end{aligned} \tag{39}$$

where the matrices are represented as

$$[A] = \begin{bmatrix} 0 & 1 & \dots & 0 & 0 \\ -\omega_{*1}^2 & 0 & \dots & 0 & 0 \\ \vdots & \vdots & \vdots & \vdots & \vdots \\ 0 & 0 & \dots & 0 & 1 \\ 0 & 0 & \dots & -\omega_{*R}^2 & 0 \end{bmatrix}$$

$$[B] = [\phi^{ss}][\beta^p] \tag{40}$$

$$[C] = [\bar{C}][\phi^{ss}]$$

and

$$[D] = [0]$$

The physical displacements and velocities at the coordinates where the encoders are attached to a structure are represented by the output vector $\{y\}$.

## 3. Evaluation Criteria

It is essential to validate the accuracy of the dynamic reduction approaches outlined in the preceding section in the context of off-road vehicles. In general, the quality of the reduced finite element models obtained by these dynamic reduction approaches characterizes their accuracy. The frequencies and mode shapes of a reduced model are typically compared to those of the full-order finite element model or reference model to determine its accuracy. An important metric for evaluating the precision of dynamic reduction techniques is the deviation of the natural frequencies from the full-order model for the reduced structure. To represent the frequency deviations between the reduced-order model and the full-order model in mathematical terms, we can use

$$\epsilon_i = \left| \frac{\omega_{*i} - \omega_i}{\omega_i} \right| \times 100\% \tag{41}$$

where $\epsilon_i$ represents the percentage error between the $i$th reduced-order and full-order mode, $i \in \{Q\}$, and where $\{Q\} = \{1, \dots, R\} \cap \{1, \dots, N\}$. For the most part, the client defines the maximum frequency deviations permitted.

The cross-orthogonality check, also known as the modal vector orthogonality check, is often used to assess the quality of reduced-order finite element models in conjunction with the frequency deviation check. In the cross-orthogonality check [74], the mass normalized modal matrix ($[\phi]$) obtained by the modal analysis is primarily employed in combination with the mass matrix ($[M_r]$) of the reduced-order model to assess the orthogonality of the modal vectors. Because reduced-order models have linearly independent degrees of freedom, each modal vector in the modal matrix should be orthogonal to the other vectors. As a result, the orthogonality relations may be expressed as

$$\{\phi_i\}^{\mathsf{T}}[M_r]\{\phi_j\} = \begin{cases} 0 : i \neq j \\ 1 : \text{otherwise} \end{cases} \tag{42}$$

Dynamic reduction techniques are based on the notion of transforming a substructure's physical coordinates into a set of generalized coordinates and then truncating modal coordinates to obtain a reduced set of equations for the system. As a consequence, exact

system solutions with reduced computational load are obtained over a limited frequency range. Modal truncation was essential in the past for structural finite element model analysis when the number of degrees of freedom exceeded the computing capability. In the last two decades, the exponential increase in processor performance, the introduction of general-purpose graphics processing units, and the availability of efficient and accurate numerical algorithms have essentially mitigated the computational constraints in the design and analysis processes of earlier times. Nonetheless, in the context of Industry 4.0, dynamic reduction methods are significant for the development of physics-based digital models for off-road vehicles. In essence, virtual models consist of the modal characteristics of physical off-road vehicles. In addition, digital twins are intended to emulate the simultaneous activities of their physical counterparts operating in an industrial environment. Consequently, solving the equation of motion in real-time that encompasses the complete set of modes constituting the virtual system is impractical. As a result, dynamic reduction-based reduced-order models are appropriate for accomplishing the requirements.

In practice, for well-formulated structural analysis problems, the lower order modes derived by dynamic reduction methods are extremely accurate, but substantial inaccuracies may occur in the higher frequency range. Furthermore, the general principle for dynamic reduction is to incorporate subcomponent modes up to 1.5–2.0 times the frequency range of interest [75], which is a value that the client specifies. However, certain subcomponents for off-road vehicles may have all the modal components exceeding the frequency range of interest. In such circumstances, the number of coordinates to be retained for each subcomponent is specified without explicitly defining the frequency range in the analysis. Subsequently, during the final system synthesis, the coordinate system obtained by synthesizing individual substructures is truncated to the desired frequency range for the overall structure.

Vibration analysis of off-road vehicles tends to focus on the lower frequency bands and the equations of motion comprised of these modes, which eventually represent their virtual models. As the number of modes incorporated into the digital model increases, so does the computational cost, which eventually deters the simulation from being synchronized with the real-time activity of its physical counterpart. Furthermore, the simulation result of digital models composed of fewer modal components may deviate from the actual behavior of the physical structure. In this research, virtual models with frequencies ranging from 0 to 30 Hz in 5 Hz steps are used to analyze the trade-off between simulation accuracy and duration.

Most textual and graphical programming platforms have built-in state-of-the-art libraries of solvers to determine the states of the explicit continuous-time state-space models given by Equation (39). It is common for these models to be simulated using variable-step solvers or fixed-step solvers, respectively. Instead of solving a model at regular intervals, the former solvers alter the step size during simulation, whereas the latter do it consistently. The adoption of fixed-step solvers for the simulation of digital models is appropriate since data are acquired from sensors connected to the physical device at fixed intervals.

During simulation, all fixed-step solvers but ode14x compute the succeeding state as

$$x(t+1) = x(t) + h\dot{x}(t) \tag{43}$$

Ode14x uses a combination of Newton's method and extrapolation from the current state to compute the succeeding state of the model, which yields

$$x(t+1) = x(t) + h\dot{x}(t+1) \tag{44}$$

The method used to evaluate $\dot{x}$ in Equations (43) and (44) is algorithm-specific and also reliant on the algorithm's order. In this research, to evaluate their efficacy, the digital models obtained by the dynamic reduction methods are solved using the following fixed-step solvers:

1. ode4 (fourth-order Runge–Kutta formula) [76];
2. ode5 (fifth-order Dormand–Prince formula) [77];
3. ode8 (eighth-order Dormand–Prince formula) [78];
4. ode14x [76].

The complexity of the algorithms rises as the order $n$ in ode$n$ increases. However, as the computational complexity grows, so does the accuracy of the result. In addition, a smaller step size enhances accuracy, but at the same time, it increases the time complexity. It is important to note that large time steps render the numerical solutions of certain models numerically unstable. With all of these factors in mind, it is essential to identify the optimal solver that achieves the trade-off between acceptable accuracy and the duration of simulation of the virtual model so that it does not exceed the operational time of its physical counterpart. To summarize, the quality of the reduced-order models obtained from model order reduction methods will be assessed using frequency deviation checks and cross-orthogonality checks. Additionally, the solutions of the digital models for several frequency ranges utilizing the stated solvers will be assessed in terms of accuracy and simulation runtime.

## 4. Results

This section presents a comprehensive assessment of the efficacy of dynamic model reduction approaches based on the evaluation methodology outlined in Section 3. The structure utilized to compare the dynamic reduction approaches is an existing commercial off-road vehicle. The structure was designed, modeled, and analyzed using the Ansys 2020R2 platform. Due to confidentiality considerations, the structure shown in Figure 1 is only a partial line diagram. The overall mass of the structure is 359 tonnes, and translational movement is restricted along the *X*- and *Z*-axis, while travel along the *Y*-axis is unrestricted. The built-in libraries of the dynamic reduction methods outlined in Section 2 are available in Ansys 2020R2, and [79,80] provides a detailed description of the procedure to develop reduced-order models from full-order finite element models using these libraries.

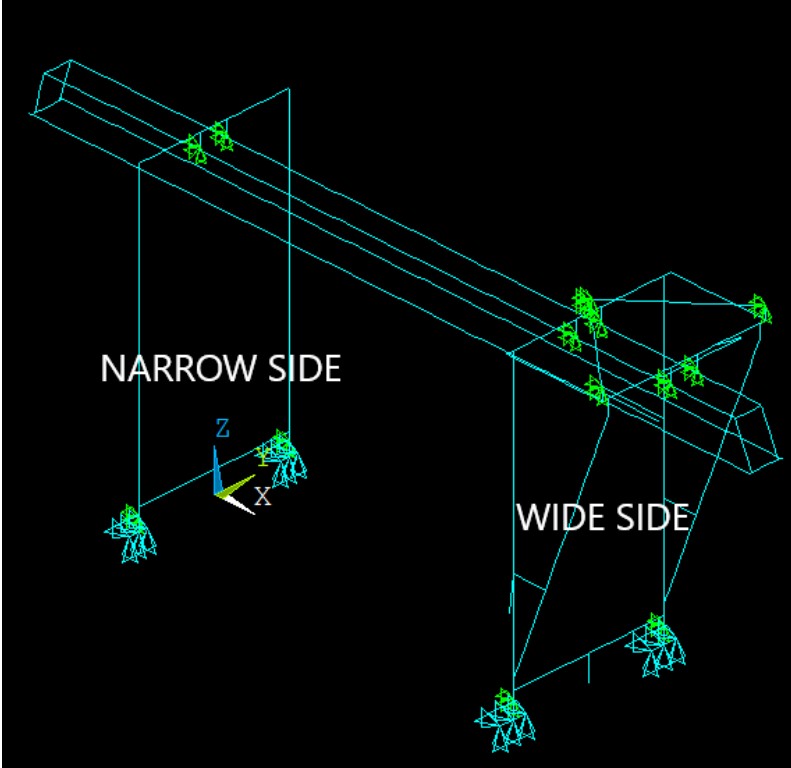

**Figure 1.** Line Diagram with Boundary Conditions and Couplings for the Structure.

Table 1 shows a comparison of the modes up to 30 Hz between full-order and reduced-order models obtained by dynamic reduction techniques.

**Table 1.** Full-Order and Reduced-Order Modes.

| Mode No. | FULL-ORDER | FREE-INTERFACE CMS | FIXED-INTERFACE CMS |
|:---:|:---:|:---:|:---:|
| | Freq (Hz) | Freq (Hz) | Freq (Hz) |
| 1 | 0.0000 | 0.0000 | 0.0000 |
| 2 | 0.1857 | 0.1857 | 0.1857 |
| 3 | 0.4302 | 0.4302 | 0.4302 |
| 4 | 1.0493 | 1.0492 | 1.0492 |
| 5 | 1.1142 | 1.1142 | 1.1142 |
| 6 | 1.8422 | 1.8414 | 1.8414 |
| 7 | 2.0013 | 2.0011 | 2.0011 |
| 8 | 2.2746 | 2.2712 | 2.2712 |
| 9 | 2.4619 | 2.4612 | 2.4612 |
| 10 | 2.9458 | 2.9453 | 2.9453 |
| 11 | 3.8005 | 3.7995 | 3.7995 |
| 12 | 4.0081 | 4.0072 | 4.0072 |
| 13 | 4.2623 | 4.2613 | 4.2613 |
| 14 | 4.4271 | 4.4151 | 4.4150 |
| 15 | 4.8720 | 4.8518 | 4.8517 |
| 16 | 5.0604 | 5.0485 | 5.0485 |
| 17 | 5.1830 | 5.1753 | 5.1753 |
| 18 | 5.6314 | 5.5985 | 5.5986 |
| 19 | 5.7793 | 5.7758 | 5.7758 |
| 20 | 5.9551 | 5.9517 | 5.9518 |
| 21 | 6.1300 | 6.1293 | 6.1292 |
| 22 | 6.2820 | 6.2816 | 6.2816 |
| 23 | 6.5562 | 6.5531 | 6.5531 |
| 24 | 6.9902 | 6.9806 | 6.9806 |
| 25 | 7.2392 | 7.2325 | 7.2325 |
| 26 | 7.4208 | 7.4186 | 7.4186 |
| 27 | 7.5028 | 7.5020 | 7.5020 |
| 28 | 7.6217 | 7.6175 | 7.6175 |
| 29 | 7.7598 | 7.7588 | 7.7588 |
| 30 | 7.9304 | 7.9310 | 7.9310 |
| 31 | 7.9935 | 7.9813 | 7.9813 |
| 32 | 8.0515 | 8.0427 | 8.0427 |
| 33 | 8.3542 | 8.3501 | 8.3502 |
| 34 | 8.4016 | 8.3975 | 8.3977 |
| 35 | 8.5372 | 8.5362 | 8.5363 |

**Table 1.** *Cont.*

| Mode No. | FULL-ORDER | FREE-INTERFACE CMS | FIXED-INTERFACE CMS |
|---|---|---|---|
| | Freq (Hz) | Freq (Hz) | Freq (Hz) |
| 36 | 8.5915 | 8.5831 | 8.5831 |
| 37 | 8.6763 | 8.6649 | 8.6649 |
| 38 | 8.8309 | 8.8182 | 8.8182 |
| 39 | 9.0159 | 9.0136 | 9.0137 |
| 40 | 9.0726 | 9.0656 | 9.0655 |
| 41 | 9.1020 | 9.1078 | 9.1078 |
| 42 | 9.3330 | 9.3323 | 9.3324 |
| 43 | 9.7096 | 9.7072 | 9.7074 |
| 44 | 9.7463 | 9.7372 | 9.7374 |
| 45 | 10.0290 | 10.0262 | 10.0263 |
| 46 | 10.1870 | 10.1680 | 10.1681 |
| 47 | 10.4320 | 10.4292 | 10.4296 |
| 48 | 11.0240 | 11.0229 | 11.0231 |
| 49 | 11.9730 | 11.9713 | 11.9714 |
| 50 | 11.9930 | 11.9919 | 11.9920 |
| 51 | 12.0290 | 12.0277 | 12.0277 |
| 52 | 12.1350 | 12.1325 | 12.1326 |
| 53 | 12.4540 | 12.4287 | 12.4288 |
| 54 | 13.2190 | 13.2155 | 13.2162 |
| 55 | 13.5810 | 13.5752 | 13.5756 |
| 56 | 13.8690 | 13.8652 | 13.8666 |
| 57 | 14.2990 | 14.2898 | 14.2900 |
| 58 | 14.7240 | 14.6956 | 14.6957 |
| 59 | 14.8700 | 14.8130 | 14.8132 |
| 60 | 15.2040 | 15.1943 | 15.1942 |
| 61 | 15.3810 | 15.3788 | 15.3792 |
| 62 | 16.6140 | 16.6105 | 16.6110 |
| 63 | 16.6540 | 16.6420 | 16.6415 |
| 64 | 16.7850 | 16.7810 | 16.7815 |
| 65 | 17.0350 | 17.0039 | 17.0039 |
| 66 | 17.1120 | 17.1068 | 17.1078 |
| 67 | 17.2230 | 17.2216 | 17.2212 |
| 68 | 17.8190 | 17.8141 | 17.8161 |
| 69 | 18.3840 | 18.3698 | 18.3721 |
| 70 | 18.7200 | 18.7067 | 18.7073 |
| 71 | 18.8380 | 18.8216 | 18.8219 |
| 72 | 18.9780 | 18.9641 | 18.9645 |
| 73 | 19.4900 | 19.4869 | 19.4882 |

**Table 1.** *Cont.*

| Mode No. | FULL-ORDER | FREE-INTERFACE CMS | FIXED-INTERFACE CMS |
|---|---|---|---|
| | Freq (Hz) | Freq (Hz) | Freq (Hz) |
| 74 | 19.5310 | 19.5261 | 19.5267 |
| 75 | 19.7960 | 19.7869 | 19.7893 |
| 76 | 19.9100 | 19.8781 | 19.8793 |
| 77 | 20.3080 | 20.3112 | 20.3124 |
| 78 | 20.7510 | 20.7381 | 20.7436 |
| 79 | 21.1300 | 21.1500 | 21.1504 |
| 80 | 21.3580 | 21.3469 | 21.3482 |
| 81 | 21.5700 | 21.5492 | 21.5493 |
| 82 | 21.7170 | 21.6348 | 21.6353 |
| 83 | 21.8040 | 21.8018 | 21.8019 |
| 84 | 22.0530 | 22.0131 | 22.0144 |
| 85 | 22.3150 | 22.3055 | 22.3086 |
| 86 | 22.4830 | 22.4724 | 22.4725 |
| 87 | 22.7500 | 22.6872 | 22.6877 |
| 88 | 23.2110 | 23.1199 | 23.1202 |
| 89 | 23.3010 | 23.2927 | 23.2955 |
| 90 | 23.3880 | 23.3803 | 23.3822 |
| 91 | 24.0510 | 23.9223 | 23.9285 |
| 92 | 24.2010 | 24.1674 | 24.1707 |
| 93 | 24.2800 | 24.2165 | 24.2187 |
| 94 | 24.3390 | 24.3309 | 24.3325 |
| 95 | 24.7000 | 24.5120 | 24.5120 |
| 96 | 25.0620 | 25.0344 | 25.0341 |
| 97 | 25.1270 | 25.0614 | 25.0636 |
| 98 | 25.3970 | 25.3898 | 25.3908 |
| 99 | 25.4890 | 25.5556 | 25.5579 |
| 100 | 25.7190 | 25.7176 | 25.7208 |
| 101 | 25.7810 | 25.7567 | 25.7612 |
| 102 | 26.2300 | 26.1940 | 26.1973 |
| 103 | 26.4390 | 26.4237 | 26.4268 |
| 104 | 26.4940 | 26.4457 | 26.4505 |
| 105 | 26.8210 | 26.8180 | 26.8201 |
| 106 | 26.9660 | 26.9501 | 26.9515 |
| 107 | 27.0410 | 27.0264 | 27.0275 |
| 108 | 27.1310 | 27.1217 | 27.1219 |
| 109 | 27.2620 | 27.2582 | 27.2630 |
| 110 | 27.4920 | 27.5396 | 27.5461 |
| 111 | 27.8520 | 27.8642 | 27.8690 |

**Table 1.** *Cont.*

| Mode No. | FULL-ORDER | FREE-INTERFACE CMS | FIXED-INTERFACE CMS |
|---|---|---|---|
| | Freq (Hz) | Freq (Hz) | Freq (Hz) |
| 112 | 28.2750 | 28.2502 | 28.2612 |
| 113 | 28.4250 | 28.4160 | 28.4172 |
| 114 | 28.7100 | 28.7304 | 28.7353 |
| 115 | 28.9110 | 28.9312 | 28.9342 |
| 116 | 29.0080 | 29.0032 | 29.0053 |
| 117 | 29.3980 | 29.4209 | 29.4278 |
| 118 | 29.6380 | 29.6299 | 29.6304 |
| 119 | 29.6620 | 29.6699 | 29.6710 |
| 120 | 29.9190 | 29.8785 | 29.8789 |

Figure 2 depicts that the frequency deviation of the reduced-order model from the full-order finite element model for both free-interface and fixed-interface modal synthesis is less than 0.8% over a frequency range of 0–30 Hz.

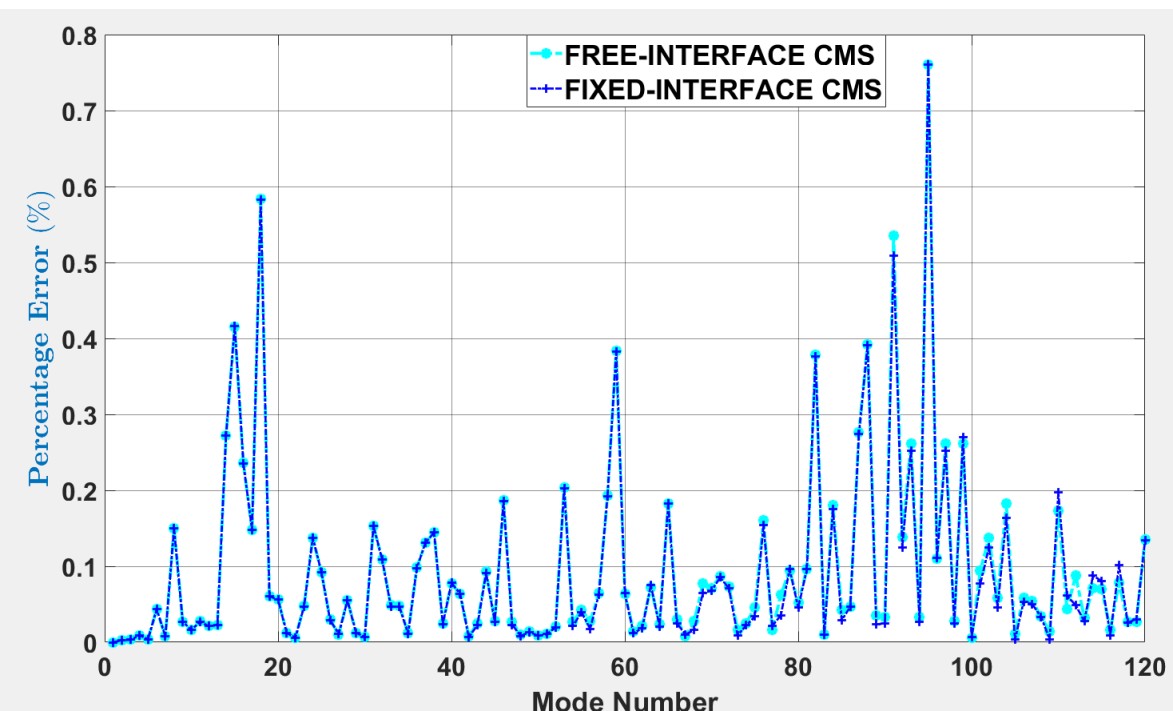

**Figure 2.** Percentage deviations of modes.

Figure 3 shows the modal vector orthogonality verification of the component modal synthesis-based reduced-order model for modes up to 30 Hz. The diagonal terms are unity, but all off-diagonal terms are near to zero for the reduced-order models obtained from both fixed-interface and free-interface component modal synthesis, as shown in the figure, which is consistent with the orthogonality relations specified in Equation (42).

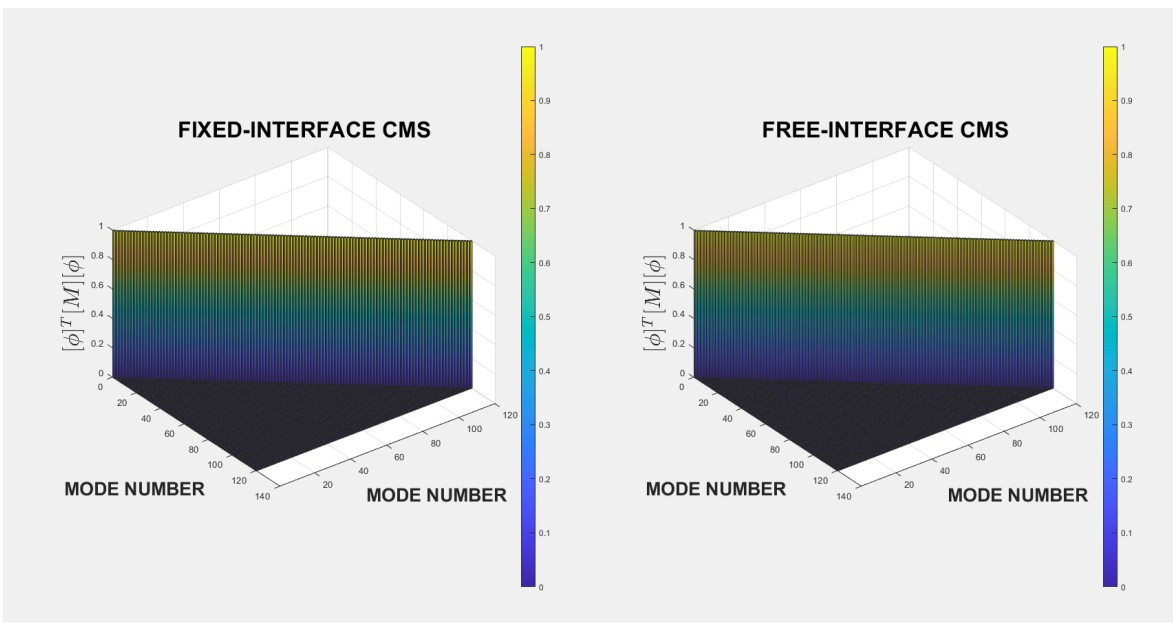

**Figure 3.** Modal Vector Orthogonality Checks of the Reduced-Order Models.

Figure 4 illustrates the digital model of the structure created utilizing either of the modal synthesis methods and integrating the corresponding driveline, electrical, and powertrain control modules in the MATLAB Simulink environment.

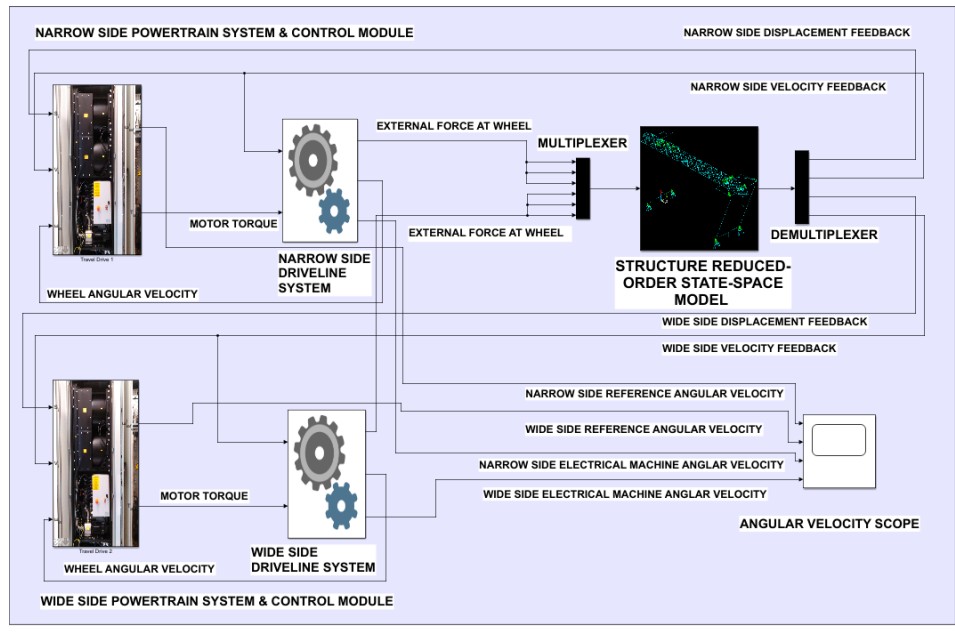

**Figure 4.** Simulation Model for the Structure in MATLAB.

The structure has twelve rigid wheels with a 710 mm diameter, three of which are attached to each of its four corners. Additionally, the structure is driven by six electrical machines, three on each side, which means that three wheels on each side are driven by electrical machines while the other three are undriven. The rated power of each electrical machine is 165 kW, the rated speed is 1413 rpm, and the rated frequency is 60 Hz. An encoder is attached to one of the driving wheels on either side. The driveline system also includes gears with a gear ratio of 26.28. The powertrain control module is composed of a proportional-integral controller (PI controller) with gain parameters ($k_P$ and $k_I$) that are a linear, monotonically decreasing function of the wheel's instantaneous angular velocity in the acceleration regime and a constant in the steady-state regime, and a linear, monotoni-

cally increasing function of the wheel's instantaneous angular velocity in the deceleration regime. As coded in the control module, the piecewise reference trapezoidal trajectory is as follows:

$$\Omega(t_i, \Omega_r, v, s) = \begin{cases} \Theta_a t_i^{as} & : t_i = t_i^{as}, s_i < s_d, v_i < v_m, \\ & \quad \Omega(t_i^{as}) < \Omega_r \\ \Omega_r & : t_i = t_i^{as}, s_i < s_d, v_i < v_m, \\ & \quad \Omega(t_i^{as}) = \Omega_r \\ \Omega_{t_{i-1}^{as}} & : t_i = t_i^{as}, s_i < s_d, v_i = v_m \\ \Theta_d t_i^d & : t_i = t_i^d, s_i \geq s_d \end{cases} \tag{45}$$

The simulation scenario is described as follows: the structure is allowed to travel a distance of 30 m along the Y-axis; subsequently, the braking regime is initiated, where the electrical machines reverse the direction of rotation; and ultimately, the simulation terminates as soon as the wheels come to a standstill. The structure is permitted to attain a maximum velocity of 2 m/s at any instant in time. The control modules, as illustrated in Figure 4, monitor both displacement and velocity at the output ports of the reduced-order model's state-space model representation through a feedback loop. In practice, the control module monitors the encoder signals for displacement and velocity.

Initially, a fixed-interface cms-based reduced-order model with modes up to 30 Hz is used to explain the dynamics of the structure, followed by a comparison of the simulation results of the reduced-order models obtained from both dynamic reduction methods. Figure 5 illustrates the simulation outcomes for the wheel's angular velocity and displacement for both the structure's narrow and wide sides. As evident from Figure 5, the braking regime commences as soon as the structure traverses 30 m, and eventually the wheels come to a complete halt at 25.5 s.

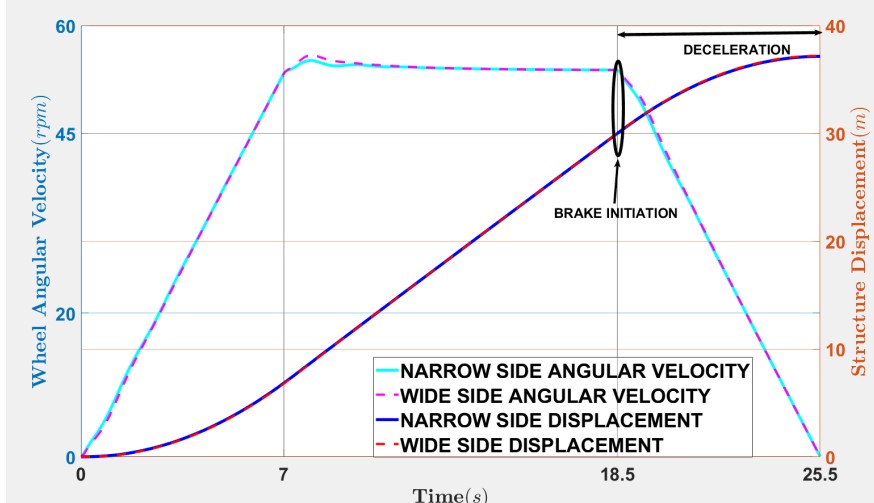

**Figure 5.** Wheel Angular Velocity and Displacement for the Structure.

In Figure 6, the simulation results for the motor angular velocities and translational velocities at the narrow and wide sides are depicted. Additionally, Figure 6 depicts the acceleration phase, which lasts until the structure reaches its maximum velocity of 2 m/s, at which point it maintains a constant velocity until the braking process begins.

Figure 7 depicts the torque at the gear output side for both narrow and wide sides, the angular velocities of the electrical machines on the narrow and wide sides, and the reference trajectory of the angular velocity that the structure is expected to track. In addition, Figure 7 shows the torque necessary to keep the angular velocities consistent with the reference ramp.

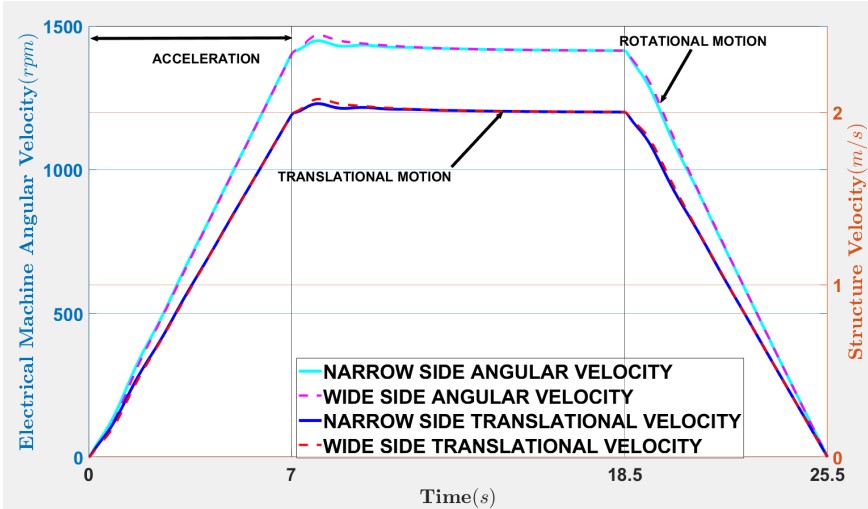

**Figure 6.** Electrical Machine Angular Velocity and Linear Velocity of the Structure.

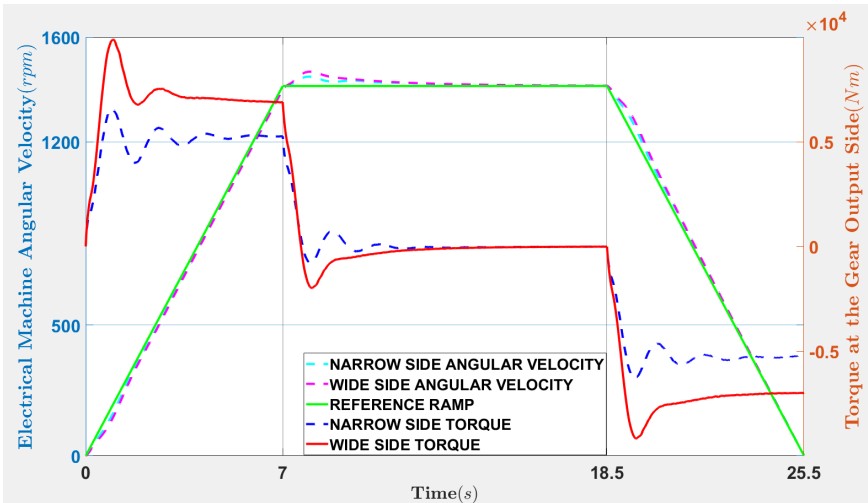

**Figure 7.** Electrical Machine Angular Velocity and Torque at the Gear Output Side.

The minimum and maximum coordinates spanned by the structure along the $X$, $Y$, and $Z$-axes are $(-19.4, 63.5)$, $(-10.14, 10.14)$, and $(0, 35.88)$, respectively, whereas the center of mass $(X, Y, Z)$ is at $(28.317, -0.417, 21.269)$, where the units are in the SI system. The minimum and maximum span of the coordinates along the $X$-axis are on the narrow and wide sides, respectively. As a result, the structure's mass is asymmetrically distributed towards the wide sides along the $X$-axis, the horizontal axis orthogonal to the direction of travel. As a direct consequence of the skewed mass distribution along the $X$-axis, there is a greater demand for torque from the wide side motors than from the narrow side motors. The simulation result shown in Figure 7 corroborates the justification. Furthermore, owing to asymmetrical mass distribution, the instantaneous velocities of the narrow and wide sides often differ during travel, as shown by the simulation result in Figure 5. Figure 8 illustrates the geometric deformation of the structure as a consequence of the mass imbalance and application specifics. In other words, if $s_N(t)$ and $s_W(t)$ are the displacements at the narrow and wide sides, respectively, at any simulation epoch $t$, then the difference in displacement at that instance is given by $(s_N(t) - s_W(t))$, and $(s_N(t) - s_W(t)) \neq 0$ indicates geometric distortion in the structure.

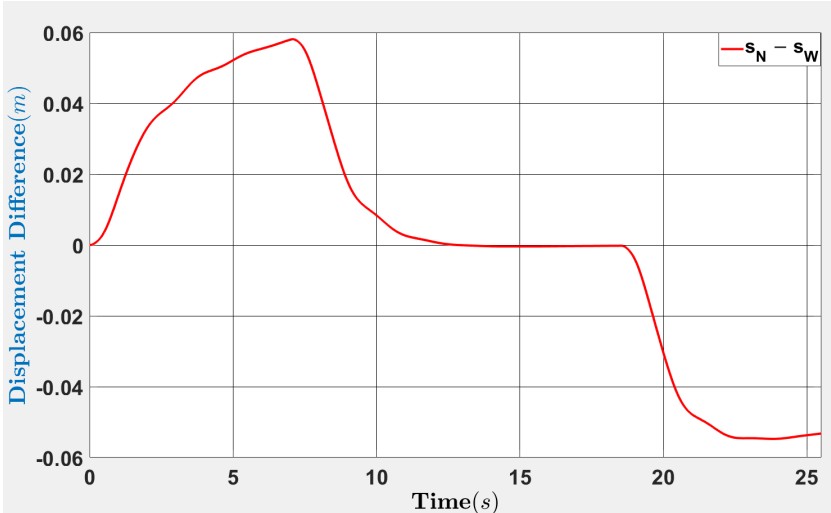

**Figure 8.** Dissimilar Displacements at Narrow and Wide Sides.

Fixed-interface component modal synthesis-based reduced-order models constituting modes up to 30 Hz have been used in the simulations thus far. Figure 9 exhibits the simulation results of the angular velocity for the electrical machine at the wide side for reduced-order models with modes up to 5 Hz, 10 Hz, and 30 Hz. The simulation result for the ROM with modes up to 5 Hz differs from the results of the other two ROMs, while there is no substantial variance between the simulation results of the ROMs with frequency ranges of 10 Hz and 30 Hz.

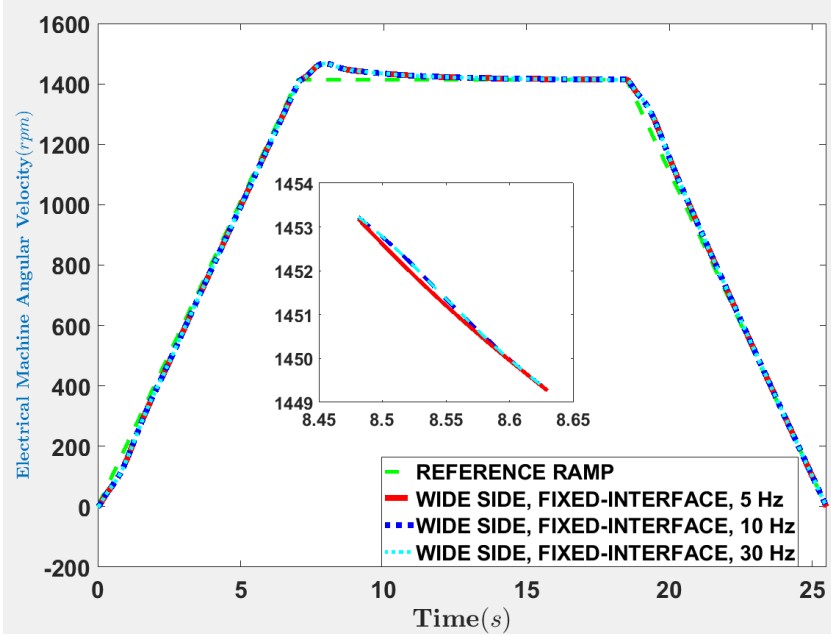

**Figure 9.** Wide Side Electrical Machine Angular Velocity for Fixed-Interface ROMs with Varying Modes.

Reduced-order models based on free-interface component modal synthesis yielded the same simulation results. Both fixed-interface component modal synthesis and free-interface component modal synthesis-based ROMs with modes ranging from 10 Hz to 30 Hz had almost indistinguishable simulation outcomes. Furthermore, there is no significant difference in simulation outcomes between ROMs based on fixed-interface and free-interface component modal synthesis methods. It is conceivable that the similarities in simulation

outcomes are attributable to the fact that the reduced-order models developed using both approaches have almost identical modes.

To this point, comprehensive findings of the outcomes for various simulation scenarios for reduced-order models based on dynamic reduction techniques have been provided. The remainder of this section concentrates on the elapsed time for the solvers to execute the simulation models, which is one of the performance metrics to identify the appropriate dynamic reduction method. A computer with an Intel Core $i7 - 8700$ CPU and 16 GB of RAM was utilized to perform all numerical simulations.

Given that the primary objective of the reduced-order model-based virtual model is to emulate the physical counterpart, a solver with a runtime longer than the physical counterpart's real-time activity is inappropriate. Figure 10 shows the time it takes to execute various numerical solvers to simulate digital models with varying frequency ranges. To be precise, the elapsed periods shown in Figure 10 are the averages of the elapsed times for 25 recurring instances. The time step used for all the fixed-step solvers is $1 \times 10^{-4}$.

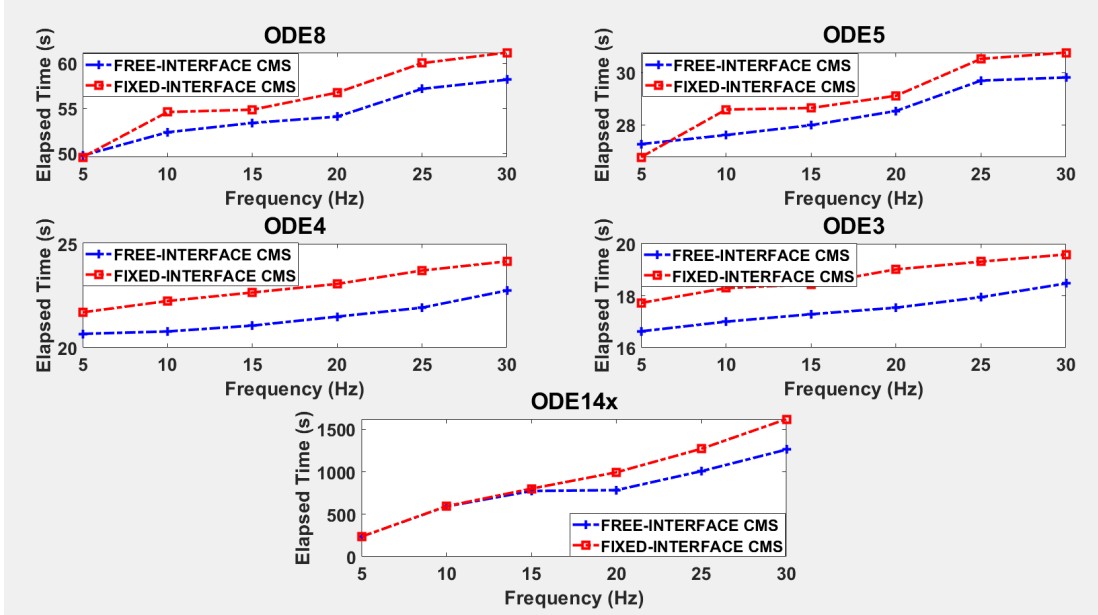

**Figure 10.** Elapsed time for various frequency ranges of digital models.

The elapsed time to numerically solve the digital models is a monotonically increasing function of the frequency ranges, as seen in Figure 10. As depicted in Figure 10 , ode8, ode5, and ode14x have runtimes that exceed 25.5 s, the upper bound for admissible simulation time, and so are inadequate to satisfy the metrics. Ode4 outperforms ode8, ode5, and ode14x in terms of execution time. However, ode3, which employs the Bogacki-Shampine formula, is the most economical approach for numerically solving digital models of off-road vehicles in terms of elapsed time. Figure 10 further illustrates that in most frequency ranges, digital models represented by free-interface component modal synthesis require less time to execute than digital models based on fixed-interface component modal synthesis for most solvers. Although there is no substantial difference in accuracy between fixed and free-interface component modal synthesis-based digital models, free-interface component modal synthesis-based digital models for off-road vehicles are deemed appropriate in a computing-resource-constrained setting.

## 5. Conclusions

The development of digital twins for off-road vehicles is at an emerging stage. There are several impediments to developing and executing virtual models for off-road vehicles. Firstly, the manual generation of off-road vehicle virtual models is inconvenient due to the sheer number and diversity of existing and operational off-road vehicles. Furthermore, de-

signing custom electrical, powertrain, and control components on an FEA-based platform is complicated. Improvements to the modeling and simulation capabilities of FEA-based software are certainly plausible, but such procedures for upgrade are challenging. Although the uncluttered syntax renders modeling the control and powertrain systems straightforward, structural design and analysis are complicated in problem-solving environments. This article proposes a method for developing digital models by using reduced-order models of existing structures on the same design platform rather than re-modeling them on a different platform, resulting in no significant additional effort. The development of reduced-order models from full-order structural finite-element models and subsequently representing the reduced-order models in state-space representation can be automated, enabling the development of structural virtual models with minimal additional effort and time.

The main contributions of the article are:

- A comprehensive review of the dynamic reduction methods, the libraries of which are available as built-in packages on most FEA-based platforms. The dynamic reduction methods eventually facilitate the development of the reduced-order models of the existing and operating structures.
- A mathematical derivation of the state-space representation of the reduced-order models.
- Establishing the performance metrics for evaluating dynamic reduction methods.
- Identifying the most appropriate dynamic reduction approach for developing digital models for off-road vehicles.
- A comparison of the numerical solvers in the problem-solving platforms.
- Selection of the optimal numerical solver to simulate the digital models for off-road vehicles.
- Identifying the lower bound of the frequency range is necessary and sufficient for developing reduced-order models for off-road vehicles.

The following are the rationale for state-space representation of the reduced-order models:

- Most commercial textual and graphical programming platforms have built-in blocks to represent the state-space models.
- It facilitates the modeling of virtual models based on ROMs incorporating structural damping.
- While performing industrial operations, the configuration of specific components within a structure varies, modifying the full-order finite element model, which subsequently alters the modal characteristics of the overall structure. As a consequence, the reduced-order model changes as well. In these scenarios, the time-varying state-space model can be used on textual and graphical programming platforms to simulate digital models based on changing ROMs.

The simulation results demonstrate that the outputs of digital models based on free-interface and fixed-interface component modal synthesis are not substantially different. However, owing to the reduced computational overload, virtual models based on free-interface component modal synthesis are more appropriate in resource-constrained settings.

**Author Contributions:** Conceptualization, S.M.; Methodology, S.M.; Software, S.M.; Validation, S.M.; Formal Analysis, S.M.; Investigation, S.M.; Resources, S.M.; Data Curation, S.M.; Writing—Original Draft Preparation, S.M.; Writing—Review and Editing, S.M., D.R. and J.W.; Visualization, S.M.; Supervision, D.R. and J.W.; Project Administration, S.M., D.R. and J.W.; Funding Acquisition, D.R. and J.W. All authors have read and agreed to the published version of the manuscript.

**Funding:** The authors would like to thank Munster Technological University for providing the opportunity to undertake this research. This work was supported, in part, by Science Foundation Ireland grant $13/RC/2094\_P2$ and co-funded under the European Regional Development Fund through the Southern & Eastern Regional Operational Programme to Lero—the Science Foundation Ireland Research Centre for Software (www.lero.ie).

**Institutional Review Board Statement:** Not applicable.

**Informed Consent Statement:** Not applicable.

**Data Availability Statement:** Not applicable.

**Acknowledgments:** This paper and the research behind it would not have been possible without the support of the IMaR team in the Munster Technological University.

**Conflicts of Interest:** The authors declare no conflict of interest.

## List of Symbols

| | |
|---|---|
| $[\bar{\phi}_n]$ | Matrix of retained fixed-constraint normal modeshapes |
| $[\bar{\phi}_{Bn}]$ | Matrix of retained free-interface modeshapes corresponding to boundary degrees of freedom |
| $[\bar{\phi}_{In}]$ | Matrix of retained free-interface modeshapes corresponding to interior degrees of freedom |
| $[\bar{C}]$ | Boolean matrix |
| $[\bar{k}]$ | Stiffness matrix in generalized coordinates |
| $[\bar{m}]$ | Mass matrix in generalized coordinates |
| $[\beta]$ | Compatibility matrix |
| $[\beta^p]$ | Boolean matrix |
| $[\phi]$ | Matrix of modeshapes for the reduced-order model |
| $[\phi^{ss}]$ | Modified modeshape matrix of reduced system |
| $[\phi_a]$ | Matrix of retained attachment modeshapes |
| $[\phi_c]$ | Matrix of constraint modeshapes |
| $[\phi_e]$ | Matrix of elastic modeshapes |
| $[\phi_n]$ | Matrix of fixed-constraint normal modeshapes |
| $[\phi_o]$ | Matrix of rigid body modeshapes |
| $[\Psi]$ | Inertia-relief matrix |
| $[A]$ | State matrix |
| $[B]$ | Input matrix |
| $[C]$ | Output matrix |
| $[D]$ | Feed-forward matrix |
| $[k]_i$ | Stiffness matrix of the $i$th substructure in physical coordinates |
| $[K_r]$ | Reduced stiffness matrix in physical coordinates |
| $[m]_i$ | Mass matrix of the $i$th substructure in physical coordinates |
| $[M_r]$ | Reduced mass matrix in physical coordinates |
| $[T_{CB}]$ | Craig–Bampton transformation matrix |
| $[T_H]$ | Hintz's transformation matrix |
| $\epsilon_i$ | Percentage error between $i$th reduced-order and full-order mode |
| $\omega$ | Natural frequencies of full-order system |
| $\Omega(t_i^d)$ | Reference trajectory at time $t_i^d$ in the deceleration regime, and where $t_0^d$ indicates the beginning of the braking of the regime |
| $\Omega(t_i^{as})$ | Reference trajectory at time $t_i^{as}$ in the acceleration and steady state regime, and where $t_0^{as}$ indicates the vehicle is commencing travel |
| $\omega_*$ | Retained modes in reduced system |
| $\omega_n$ | Fixed-constraint normal modes |
| $\Omega_r$ | Electrical machine rated speed |
| $\omega_{fn}$ | Free-interface normal modes |
| $\tau]$ | Total number of substructures or components |
| $\tau_c$ | Total number of couplings in the structure |
| $\Theta_a$ | A constant and $\Theta_a > 0$ |
| $\Theta_d$ | A constant and $\Theta_d < 0$ |
| $\{\bar{u}\}$ | Global physical displacements |
| $\{\bar{u}_B\}$ | Global physical displacements for uncoupled boundary coordinates |
| $\{\ddot{u}\}$ | Local acceleration in physical coordinates |
| $\{\eta\}$ | Modal coordinates |

| $\{\eta_r\}$ | Modal coordinates for reduced-order model |
| $\{\mu\}$ | Generalized coordinates |
| $\{\phi_B\}$ | Eigenvectors of free-interface modes corresponding to boundary degrees of freedom |
| $\{\phi_I\}$ | Eigenvectors of free-interface modes corresponding to interior degrees of freedom |
| $\{p\}$ | External forces in physical coordinates |
| $\{p_o\}$ | D'Alembert's forces or inertial forces |
| $\{p_r\}$ | External force at the $r$th degree of freedom of the reduced-order model |
| $\{u\}$ | Local displacements in physical coordinates |
| $\{u_e\}$ | Elastic deformation vector |
| $\{u_o\}$ | Rigid body displacement |
| $\{x\}$ | State variables |
| $\{z\}$ | Input vector |
| $B$ | Boundary degrees of freedom in $i$th substructure |
| $h$ | Simulation step size |
| $I$ | Interior degrees of freedom in $i$th substructure |
| $N$ | Total degrees of freedom in full-order system |
| $N_i^I$ | Total degrees of freedom for $i$th substructure |
| $R$ | Retained degrees of freedom in reduced system |
| $s_d$ | Distance to be traveled |
| $s_i$ | Displacement of the vehicle at time $t_i^{as}$ or $t_i^d$ |
| $s_N$ | Displacement at the narrow side |
| $s_W$ | Displacement at the wide side |
| $t$ | Simulation epoch |
| $v_i$ | Velocity of the vehicle at time $t_i^{as}$ or $t_i^d$ |
| $v_m$ | Maximum allowable velocity for the off-road vehicle |

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
