# Peer review of "Dynamic Reduction-Based Virtual Models for Digital Twins—A Comparative Study"

_applsci, doi:10.3390/app12147154_

Round 1
Reviewer 1 Report
Dear Authors,
while i was reading your work i had the following concerns:
A digital Twin is a much bigger topic then only an accurate simulation of an physical domain.
There are many other things according to data (ingestion, aggregation, visualization and standardization), the virtual model (simulation, prediction) and usability (HMI design, interaction, virtual interaction).
See "https://unity.com/solutions/digital-twin" which is roughly fitting to our interpretation of a digital twin.
Therefore, your title seems for me a bit irritating, since you do not address such topics.
Section 2 seems for me as an collection of the state of the Art.
If it is so, then the portion if this chapter is to big, according to the whole work (takes nearly half of the contribution).
If not, you should highlight your work in these sections better.
Further, you have nearly no reference to this section, which makes it feel like the basic background each reader should know.
So questions arise like: Why not just reference other work and give instead a short summary so that the following work can be understood?
Your own work seems for me to start at section 4, where you create the model with Ansys.
In this chapter i miss how you created your reduced model.
You directly compare the Ansys model with the reduced one.
Where does it come from?
Your simulation model (Fig. 5) seems not to follow any convention.
It shows the connection of the entities and has rough descriptions at some edges, but not all edges have an description and not all entities are defined (e.g. the Box with 4 Inputs at the bottom right corner).
In general i could understand what you have done.
But i was questioning myself, how your work differs from classical modal analysis and model order reduction.
If you could highlight this stronger, it would be much clearer for the reader where your work starts.
I hope this will help during paper improvement.
Best regards!
Author Response
Thank you for taking the time out to read the article. We appreciate your suggestions. The amended document is now available for review. The text deleted from the original article is displayed in red, while the content inserted into the revised version is represented in blue.
Comment 1: A digital Twin is a much bigger topic then only an accurate simulation of an physical domain.
There are many other things according to data (ingestion, aggregation, visualization and standardization),
the virtual model (simulation, prediction) and usability (HMI design, interaction, virtual interaction).
See https://unity.com/solutions/digital-twin which is roughly fitting to our interpretation of a digital twin.
Therefore, your title seems for me a bit irritating, since you do not address such topics.
Response: Thank you for bringing this to our attention. We have modified the title to "Dynamic Reduction-based Virtual Models for Digital Twins-A Comparative Study".
Comment 2: Section 2 seems for me as an collection of the state of the Art.
If it is so, then the portion if this chapter is to big, according to the whole work
(takes nearly half of the contribution).
If not, you should highlight your work in these sections better.
Further, you have nearly no reference to this section, which makes it feel like the basic background each
reader should know.
So questions arise like: Why not just reference other work and give instead a short summary so that the following
work can be understood?
Response: Sections 2.1 and 2.2 are reviews of Roy Craig and Mervyn Bampton's "Coupling of Substructures for Dynamic Analyses" and Robert Morris Hintz's "Analytical Methods in Component Modal Synthesis", respectively. Most FEA-based software packages include built-in libraries for these dynamic reduction methods. These libraries predominantly use terminologies not explicitly mentioned in the original articles. Ansys, for example, utilizes the term "Master DOFs" to represent the degrees of freedom to be retained in the reduced-order model of a substructure or component. In addition, some of these Master Coordinates are "coupled points" while others are employed to apply external inputs. Reviews of dynamic reduction methods presented in these sections provide these practical aspects of the algorithms while preserving the original article's fundamentals. We hope this will help readers connect the theoretical foundation with the practical techniques for developing reduced-order models for off-road vehicles. It is worth noting that since the FEA-based platforms use different notations and terminologies, this article makes no explicit reference to any platform-specific nomenclature. The concept and mathematical derivation of the state-space representation of the reduced-order models presented in Section 2.3 are novel contributions of the authors. In both the Introduction and the Conclusion, the authors' contributions are mentioned. Section 2 has been updated to include the references.
Comment 3: Your own work seems for me to start at section 4, where you create the model with Ansys.
In this chapter i miss how you created your reduced model.
You directly compare the Ansys model with the reduced one.
Where does it come from?
Response: The Ansys manuals (references 79 and 80), which outline the procedures for utilizing the libraries about component modal synthesis, have been incorporated as references.
Comment4: Your simulation model (Fig. 5) seems not to follow any convention. It shows the connection of the entities and has rough descriptions at some edges, but not all edges have an description and not all entities are defined (e.g. the Box with 4 Inputs at the bottom right corner).
Response: Figure 4 now has all of its blocks and signals labeled.
Comment 5: But i was questioning myself, how your work differs from classical modal analysis and model order reduction.
If you could highlight this stronger, it would be much clearer for the reader where your work starts.
Response: The main contributions of the article are:
- A comprehensive review of the dynamic reduction methods, the libraries of which are available as built-in packages on most FEA-based platforms. The dynamic reduction methods eventually facilitate the development of the reduced-order models of the existing and operating structures.
- A mathematical derivation of the state-space representation of the reduced-order models.
- Establishing the performance metrics for evaluating dynamic reduction methods.
- Identifying the most appropriate dynamic reduction approach for developing digital models for off-road vehicles.
- A comparison of the numerical solvers in the problem-solving platforms.
- Selection of the optimal numerical solver to simulate the digital models for off-road vehicles.
- Identifying the lower bound of the frequency range is necessary and sufficient for developing reduced-order models for off-road vehicles.
The preceding paragraph is a portion of the article's "Conclusion", which explicitly mentions the authors' contributions.
Furthermore, the following content has been incorporated into the revised article's introductory section:
The virtual models facilitate the yield of physics-based digital twins for off-road vehicles; however, there is hardly any extant literature presenting their development methods. This paper proposes dynamic reduction methods for developing reduced-order models that subsequently represent the virtual models for off-road vehicles. Simulation of these digital models with reduced degrees of freedom improves speed without impacting accuracy. Furthermore, built-in packages of FEA-based software, predominately used for structural modeling and analysis, incorporate these high-fidelity dynamic reduction techniques, thereby enabling the development of the reduced-order models on the same platforms where designed. This article outlines the state-space representation method for these reduced-order models. The state-space models facilitate the simulation of reduced-order models in problem-solving environments. This paper, in addition, provides a comprehensive mathematical derivation for the state-space representation of reduced-order models. To accomplish real-time synchronization with the activities of a physical counterpart in an industrial environment, the simulation time of virtual models must be less than or equal to the operating time of physical equivalents. As a result, a comparative analysis of the execution times in a problem-solving environment of virtual models developed utilizing Craig-Bampton and Hintz's component modal synthesis is provided. It is worth noting that these virtual models are eventually solved utilizing solvers that are available as built-in libraries on textual and graphical programming platforms; hence, selecting the optimal solver among the available solvers is critical. Consequently, this paper presents a comparative assessment of the execution times of virtual models for off-road vehicles employing these solvers.

Reviewer 2 Report
The article is very interesting and presents the topic very clearly. The topic is very useful and still has a lot of scientific potential. Nevertheless, I suggest minor corrections that will further improve the article.
- I suggest that the authors use more newest literature. In addition to the current literature, the authors should add literature that is more recent than 2020. There are many recent articles on the topic of Industry 4.0 and digital twins.
- In the introduction, the authors should be more clear about the research gap and the scientific contribution of the research. At the current state, this is not very clear.
- In chapter 2, the authors should write how the methods relate to Industrie 4.0 and how it relates to digital twins, which they define in the first chapter.
- The authors should revise the style again - some paragraphs are staggered, others are not - they need to be unified throughout the article.
- In the results section I have some recommendations that should be followed in the final version of the article. Could the authors present the results in tabular form and not just as graphs. Figure 2 shows the different lines, but their difference is not perceptible and therefore the picture is very unclear. Would not it be useful to show only one section where the difference can be seen? The same applies to Figure 10.
Author Response
Thank you for taking the time out to read the article. We appreciate your suggestions. The amended document is now available for review. The text deleted from the original article is displayed in red, while the content inserted into the revised version is represented in blue.
Comment 1: I suggest that the authors use more newest literature. In addition to the current literature, the authors should add literature that is more recent than 2020. There are many recent articles on the topic of Industry 4.0 and digital twins.
Response: As you recommended, references to the most recent and relevant literature have been incorporated.
Comment 2: In the introduction, the authors should be more clear about the research gap and the scientific contribution of the research. At the current state, this is not very clear.
Response: The following content has been incorporated into the article's introductory section:
The virtual models facilitate the yield of physics-based digital twins for off-road vehicles; however, there is hardly any extant literature presenting their development methods. This paper proposes dynamic reduction methods for developing reduced-order models that subsequently represent the virtual models for off-road vehicles. Simulation of these digital models with reduced degrees of freedom improves speed without impacting accuracy. Furthermore, built-in packages of FEA-based software, predominately used for structural modeling and analysis, incorporate these high-fidelity dynamic reduction techniques, thereby enabling the development of the reduced-order models on the same platforms where designed. This article outlines the state-space representation method for these reduced-order models. The state-space models facilitate the simulation of reduced-order models in problem-solving environments. This paper, in addition, provides a comprehensive mathematical derivation for the state-space representation of reduced-order models. To accomplish real-time synchronization with the activities of a physical counterpart in an industrial environment, the simulation time of virtual models must be less than or equal to the operating time of physical equivalents. As a result, a comparative analysis of the execution times in a problem-solving environment of virtual models developed utilizing Craig-Bampton and Hintz's component modal synthesis is provided. It is worth noting that these virtual models are eventually solved utilizing solvers that are available as built-in libraries on textual and graphical programming platforms; hence, selecting the optimal solver among the available solvers is critical. Consequently, this paper presents a comparative assessment of the execution times of virtual models for off-road vehicles employing these solvers.
Comment 3: The authors should revise the style again - some paragraphs are staggered, others are not - they need to be unified throughout the article.
Response: We appreciate your bringing this issue to our notice. In the updated document, a standardized paragraph style has been adopted.
Comment 4: In the results section I have some recommendations that should be followed in the final version of the article.
Could the authors present the results in tabular form and not just as graphs.
Figure 2 shows the different lines, but their difference is not perceptible and therefore the picture is very unclear. Would not it be useful to show only one section where the difference can be seen? The same applies to Figure 10.
Response: Figure 2 has been replaced with a table (Table 1) illustrating the modes of the Full-Order and Reduced-Order models. \\ Figure 10 has hundreds of thousands of data-points; hence, presenting them in a tabular format is not the ideal option. Figure 10 has, however, been modified to enhance visual clarity.

Reviewer 3 Report
- Nomenclature list must be added to help readers for understanding the proposed model.
- Please add the comparison with the exiting studies for highlighting originality of the proposed method.
Author Response
Thank you for taking the time out to read the article. We appreciate your suggestions. The amended document is now available for review. The text deleted from the original article is displayed in red, while the content inserted into the revised version is represented in blue.
Comment 1: Nomenclature list must be added to help readers for understanding the proposed model.
Response: The article is updated with the list of the symbols used.
Comment 2: Please add the comparison with the existing studies for highlighting the originality of the proposed method.
Response: To the best of our knowledge, research on related topics does not exist, which precludes us from comparing our methods and models.
